# Systemic Treatment Selection for Patients with Advanced Pancreatic Neuroendocrine Tumours (PanNETs)

**DOI:** 10.3390/cancers12071988

**Published:** 2020-07-21

**Authors:** Vera G. Megdanova-Chipeva, Angela Lamarca, Alison Backen, Mairéad G. McNamara, Jorge Barriuso, Sonia Sergieva, Lilia Gocheva, Was Mansoor, Prakash Manoharan, Juan W. Valle

**Affiliations:** 1Department of Medical Oncology, The Christie NHS Foundation Trust, Manchester M204BX, UK; megdanova.vera@abv.bg (V.G.M.-C.); alison.backen@christie.nhs.uk (A.B.); Mairead.McNamara@christie.nhs.uk (M.G.M.); jorge.barriuso@manchester.ac.uk (J.B.); Was.Mansoor@christie.nhs.uk (W.M.); 2Department of Radiotherapy and Medical Oncology, University Hospital “Queen Yoanna” ISUL, 1000 Sofia, Bulgaria; lgocheva2001@yahoo.co.uk; 3Department of Nuclear Medicine, Radiotherapy and Medical Oncology, Medical University—Sofia, 1000 Sofia, Bulgaria; 4Division of Cancer Sciences, University of Manchester, Manchester M204BX, UK; 5Nuclear Medicine Department, SBALOZ, Sofia grad, 1000 Sofia, Bulgaria; sergieva.sonya@yahoo.com; 6Department of Radiology and Nuclear Medicine, The Christie NHS Foundation Trust, Manchester M204BX, UK; Prakash.Manoharan@christie.nhs.uk

**Keywords:** treatment, chemotherapy, PRRT, targeted therapy, pancreatic, somatostatin analogues, neuroendocrine

## Abstract

Pancreatic neuroendocrine tumours (PanNETs) are rare diseases and a good example of how research is not only feasible, but also of crucial importance in the scenario of rare tumours. Many clinical trials have been performed over the past two decades expanding therapeutic options for patients with advanced PanNETs. Adequate management relies on optimal selection of treatment, which may be challenging for clinicians due to the fact that multiple options of therapy are currently available. A number of therapies already exist, which are supported by data from phase III studies, including somatostatin analogues and targeted therapies (sunitinib and everolimus). In addition, chemotherapy remains an option, with temozolomide and capecitabine being one of the most popular doublets to use. Peptide receptor radionuclide therapy was successfully implemented in patients with well-differentiated gastro-entero-pancreatic neuroendocrine tumours, but with certain questions waiting to be solved for the management of PanNETs. Finally, the role of immunotherapy is still poorly understood. In this review, the data supporting current systemic treatment options for locally advanced or metastatic PanNETs are summarized. Strategies for treatment selection in patients with PanNETs based on patient, disease, or drug characteristics is provided, as well as a summary of current evidence on prognostic and predictive biomarkers. Future perspectives are discussed, focusing on current and forthcoming challenges and unmet needs of patients with these rare tumours.

## 1. Introduction 

Pancreatic neuroendocrine neoplasms (PanNENs) are rare tumours, accounting for 2–5% of pancreatic malignancies and 6–7% of all NENs, with an estimated annual incidence of 0.48 per 100,000 persons [1,2,3,4]. The median age at diagnosis is 60 years, with a slight predominance of female gender [5]. The number of patients with newly-diagnosed PanNENs is increasing (predominantly non-functional tumours), mainly due to increased awareness and improved diagnostic techniques. Pancreatic neuroendocrine neoplasms exhibit a shorter overall survival (OS) as compared to other gastro-entero-pancreatic (GEP)-NENs, with five-year OS of 38% according to the Surveillance, Epidemiology, and End Results (SEER) registry [1,5], though a more optimistic outcome has been reported in several European studies [2,6,7,8]. One of the reasons for the low survival rate is that over 50% of patients with PanNENs are diagnosed at an advanced stage (defined as locally advanced or metastatic), which is among the most important prognostic factors [9]. The extent of metastatic disease (e.g., unilobar or bilobar hepatic metastasis, sparing of extrahepatic disease), together with Ki-67, are also strong factors that influence the progression-free survival (PFS) and OS [10], together with others, such as advanced age [11]. 

Pancreatic neuroendocrine neoplasms are classified based on evidence of hormone-related symptoms, these form two groups: non-functioning (NF-PanNEN) or functioning (F-PanNEN). The latter account for a minority (30%) of all PanNENs, and may secrete hormones and peptides, such as gastrin, glucagon, insulin, and vasoactive intestinal peptide (VIP), amongst others [12]. Although the majority of PanNENs are sporadic, they may arise as part of a few hereditary conditions, like multiple endocrine neoplasia (MEN)-1 (responsible for 20–30% of gastrinomas and <5% of insulinomas), von Hippel Lindau disease, neurofibromatosis-1, and tuberous sclerosis. 

Based on pathological characteristics, PanNENs had previously been classified into grade 1 pancreatic neuroendocrine tumours (PanNETs) (well-differentiated morphology with Ki-67 < 3%); grade 2 PanNETs (well-differentiated morphology with Ki-67 3–20%) and grade 3 neuroendocrine carcinoma (Ki-67 > 20%, regardless of morphology). In 2017, the new WHO grading system for PanNENs introduced a further sub-group: well-differentiated NET with Ki67 > 20% (grade 3 PanNET), distinct from poorly-differentiated neuroendocrine carcinoma (grade 3 PanNEC) (Table 1) [13,14]. 

The goals of systemic treatment in patients with locally advanced and metastatic PanNET are to increase survival, induce tumour shrinkage, relieve symptoms, prevent tumour-related complications, and improve quality of life. The latest European Neuroendocrine Tumour Society (ENETS) guidelines, last updated in 2016, base treatment decisions on the extent of disease, grading, functionality, and the presence/absence of other symptoms not connected with hormonal production [15]. Options of systemic therapy for patients with well differentiated PanNETs include somatostatin analogues (SSAs) [16,17,18,19,20,21], targeted therapies (such as everolimus [22,23,24,25] and sunitinib [26,27]), chemotherapy (streptozotocin/5-fluorouracil (STZ/5-FU) [28,29,30] or temozolomide/capecitabine (TemCap) [31,32,33,34,35]), Peptide Receptor Radionuclide Therapy (PRRT) [36,37,38,39,40,41,42,43,44,45,46,47], and immunotherapy [48,49,50]. 

As the options of systemic therapy increase, it becomes more challenging for clinicians to select the most appropriate option or sequence of treatments for individual patients. The aim of this review is to provide a comprehensive summary of the current evidence supporting the use of systemic treatment for patients that are diagnosed with PanNETs; adequate systemic treatment selection based on patient’s individual characteristics will be discussed.

## 2. Evidence Regarding the Use of Available Systemic Treatment Option

### 2.1. Evidence Supporting the Use of Somatostatin Analogues (SSAs)

Somatostatin analogues act by targeting somatostatin receptors (SSTR 1–5) [51]. The best characterised SSAs are octreotide long-acting release (LAR) and lanreotide autogel, which mainly target SSTR-2 (expressed in about 80% of PanNETs [52]) and SSTR-5. In contrast, the next-generation SSA (pasireotide) targets a wider range of SSTRs (SSTR-1, -2, -3, and -5) [53,54]. Because of their anti-secretory effect, SSAs have been used for many years for symptom control only [51]. However, their anti-proliferative effect [55,56]) is now well established [57], Appendix A summarises the main clinical trials exploring the role of SSAs in PanNETs. 

The first robust evidence of the anti-proliferative effect of SSAs came from the PROMID clinical trial [20,21]; this prospective phase III randomised, placebo-controlled, double-blind study assessed the use of octreotide LAR in patients with locally-advanced or metastatic, treatment-naïve grade (G) 1 midgut NET, or NET with an unknown origin. Improvement of median time to tumour progression (TTP) was statistically and clinically significant (octreotide LAR 14.3 months vs. placebo 6 months, Hazard Ratio (HR) 0.34 (95%-CI 0.20–0.59; *p* = 0.000072). Patients in the placebo arm were allowed to cross over to octreotide LAR at time of progression, which is likely the main reason why the differences on TTP did not translate into OS improvement. Although patients with PanNETs were not included in the PROMID trial, the results were considered to be strong and led to the use of octreotide with anti-proliferative intent for patients with PanNETs in ENETS Guidelines [15,58]. This approach was validated in several retrospective series and small phase II study demonstrating anti-proliferative activity of octreotide LAR in PanNETs, mostly in low Ki-67 NETs (more durable responses in patients with Ki-67 < 10) [59]. 

The pivotal phase III trial assessing the effect of SSAs in patients with PanNETs was the CLARINET study [16,17,19,60]. This randomized, double-blind, placebo-controlled study evaluated lanreotide autogel in patients with locally advanced or metastatic, non-functioning (except gastrinomas), well-differentiated GEP-NETs with Ki-67 < 10%. The study period lasted for 96-weeks (core study), followed by an open label extension (OLE) part. Most of the patients were treatment-naïve (84% in both arms) and had stable disease at baseline (96% in the lanreotide and 95% in the placebo arms, respectively). The study showed a benefit in terms of PFS with a HR of 0.58 (95%-CI 0.32–1.04, core study) [16] and median PFS of 29.7 months for the group of PanNETs (whole core and OLE study) [19]; the benefit in PFS was observed regardless of tumour burden [19]. Despite the low response rate (2%), disease stabilisation was high (64%), achieving a high disease control rate (DCR) of 66%. Within the OLE part, data on patients who had crossed over to lanreotide autogel after progression on placebo and patients on lanreotide autogel with no progression at 96^th^ week (*n* = 88) was reported; interestingly, half of these were PanNETs [17]. The median PFS for patients with PanNETs was 29.7 months, being somewhat shorter than the median PFS for all patients recruited into the CLARINET trial (38.5 months) [19]. 

Many studies have aimed to increase the anti-tumour effect of SSAs, by development of new-generation SSAs such as pasireotide LAR [61] (Appendix A) or by developing combinations of SSAs with other anti-tumour agents, such as everolimus (COOPERATE-1 study [62] (Appendix A). Unfortunately, these efforts were not successful and the use of SSAs in PanNETs is currently limited to single agent strategies. 

### 2.2. Evidence Supporting the Use of Targeted Therapies

An important paradigm change arose from an improved understanding of the role of the mammalian target of rapamycin (mTOR) and angiogenesis in tumour growth and progression. Appendix A summarises the main clinical trials of the use of targeted therapies in patients with PanNETs. 

The inhibition of mTOR, with everolimus, was postulated as a promising strategy in the RADIANT-1 phase II study [22]. This led to RADIANT-3, a large phase III prospective, randomised, placebo-controlled, double-blind trial of patients with well-differentiated PanNETs who were randomised to receive everolimus or placebo [23,24,25]. Objective responses were low (<5%) and independent of prior treatment with chemotherapy. The study showed a longer median PFS with everolimus (11 vs. 4.6 months; HR 0.35; 95%-CI, 0.27–0.45; *p* < 0.001); due to cross-over, this benefit did not impact OS [23,24,25]. Target-specific side effects included hyperglycaemia, pneumonitis, infection, and stomatitis; however, G3/4 serious adverse events (SAEs) were relatively few. Alternative inhibitors of mTOR complex-1 (mTORC1) and mTOR complex-2 (mTORC2), such as BEZ235, have been tested in patients with PanNETs with disappointing results [63].

Sunitinib is a multi-tyrosine kinase inhibitor that inhibits vascular growth factor receptor (VEGFR-2 and -3), platelet derived growth factor receptor (PDGFR) and stem-cell factor receptor (c-kit). Building on the phase II study [64], the SUN111 phase III double-blind, placebo-controlled clinical trial in patient with PanNETs showed a clinically meaningful benefit of sunitinib over placebo with improved median PFS (11.4 vs. 5.5 months for sunitinib and placebo, respectively; HR 0.42; 95%-CI, 0.26–0.66; *p* < 0.001). The objective response rate (ORR) was 9%, including two patients with complete responses (CR) to sunitinib [26,27]. Longer median PFS in patients with Ki-67 < 5% (HR 0.38 (95%-CI 0.16–0.92)) and higher response rate for patients with non-functioning tumours (HR 0.26 (0.13–0.54)) were reported. No impact on OS was identified due to cross-over (38.6 vs. 29.1 months (HR 0.73; 95%-CI 0.50–1.06); *p* = 0.094)). 

Sunitinib and everolimus have been approved for treatment of PanNETs for many years now due to the results from the performed studies. Collected data regarding the use of both targeted agents in real world setting confirm their benefit with similar PFS and even higher RR on average in the phase IV and retrospective studies [10,65,66,67,68,69] as you may see on Appendix A. Everyday practice shows that although active, sunitinib and everolimus are derive on significant toxicity and grade 3/4 side effects for patients (Appendix A). 

Other antiangiogenic molecules have been tested in patients with PanNETs over the last 10 years in several phase II trials, such as pazopanib (targeting VEGFR, PDGFR, c-KIT, and fibroblast growth factor receptors (FGFR)); cabozantinib (targeting hepatocyte growth factor receptor protein (MET), VEGFR, RET, GAS6 receptor (AXL), KIT, Fms-like tyrosine kinase-3 (FLT3)); lenvatinib (targeting VEGFR, FGFR, PDGFR alpha, c-Kit, and the RET proto-oncogene); and surufatinib (targeting VEGFRs, FGFR, colony-stimulating factor 1 receptor) [18,70,71,72,73]. In all of these studies, mixed populations of patients with GEP-NETs were recruited, with limited numbers of patients with PanNETs (20–55). Most patients had already received previous targeted therapies with the aim of overcoming resistance to previous everolimus and/or sunitinib. The median PFS for patients with PanNETs ranged from 11.7 months for pazopanib [18] to 21.8 months for cabozantinib [71]. Among the expected moderate responses (between 15–19%) [18,71,73], there was a promising 42.3% response rate reported with lenvatinib [72]. With lenvatinib, the responding patients had a significantly better PFS compared to non-responders, a finding that is even more evident for the PanNET subgroup (median PFS in responders vs. non-responders: not reached (NR) vs. 11.2 months in PanNETs (*p* = 0.004). Phase III studies are now running with cabozantinib (NCT03375320) and surufatinib (NCT02589821), which will further explore the role of these novel agents in the field of PanNETs. 

Other targeted therapies have been tested, including PARP inhibitors (palbociclib [74]), novel VEGF inhibitors (ziv-aflibercept [75]), histone deacetylase (HDAC) inhibitors (panobinostat [76]), and anti-IGF1R (Insulin-like growth factor) antibodies (ganitumab (AMG 479) [77]), with disappointing results. A combination of targeted therapies with other agents (e.g., sunitinib combined with evofosfamide (a DNA alkylator bromo-isophosphoramide mustard) has led to encouraging ORRs (17.6%), but with very high levels of grade 3/4 toxicity (52.9%) [78].

### 2.3. Evidence Supporting the Use of Peptide Receptor Radionuclide Therapy (PRRT)

The ability to deliver radioactive particles, based on a diagnostic nuclear medicine scan (theragnostic concept), is the essence of PRRT. Peptide receptor radionuclide therapy consists of a chelator (DOTA) or linker that binds on one side with a SSTR ligand (agonist or antagonist) and a therapeutic particle (Yttrium-90 (^90^Y) or Lutetium-177 (^177^Lu)) on the other [43]. The NETTER-1 trial was the pivotal phase III study performed assessing the use of PRRT in patients with NETs [36]. Patients diagnosed with metastatic midgut NET, who had progressed on SSAs were recruited (patients with PanNETs were excluded from the NETTER-1 trial). Peptide receptor radionuclide therapy improved median PFS in patients with low liver tumour burden (LTB) (28.35 months with PRRT vs. 11.04 months in the control arm (HR = 0.218, 95% CI 0.120–0.394)), in patients with moderate LTB (NR vs. 8.67 (HR = 0.202, 95% CI 0.077–0.525)), and in patients with high LTB (19.38 vs. 5.52 (HR = 0.193, 95% CI 0.079–0.474) [79]. In addition, both median OS (NR vs. 27.4 months; HR 0.398, 95%-CI 0.207–0.766) and response rate (RR, 17% vs. 3%) were more favourable in the PRRT arm as compared to high dose of octreotide LAR (control arm) [36].

Unfortunately, activity data on PRRT use in patients with PanNETs are not based on randomised Phase III studies, since these patients were not included in the NETTER-1 study (Appendix A). ^90^Y and ^177^Lu use were both explored in retrospective series and phase II studies and prolonged median PFS and OS, between 20–39 months and 40.1–76 months, respectively [37,38,39,40,41,42,43,44,45,46,47] have been reported. In the majority of the studies, the RRs were high (24–54%), with few exceptions (objective response of 13% for patients with PanNETs was reported by Hamiditar et al. [45]). 

The largest series reporting outcome data in patients treated with PRRT was the study by Brabander and colleagues [80]. In this single-arm study, efficacy of ^177^Lu was explored in patients with a variety of SSTR-positive NETs; the final analysis was limited to Dutch GEP-NET population which included 197 patients with a pancreatic primary. This study showed high response rates (ORR 60.9%, 95%-CI 52.1–69.2%), long median PFS (30.5 months) and median OS (70.8 months) that were even more favourable in patients with PanNETs with progressive disease at baseline (median PFS 35.6 months; median OS 80.7 months) [80]. The magnitude of effect (survival) was similar to that seen in the NETTER-1 study for patients with intestinal NETs, leading to the approval of PRRT for the treatment of patients with PanNETs [80]. 

### 2.4. Evidence Supporting the Use of Chemotherapy

Chemotherapy has been a therapeutic option for patients with well-differentiated PanNETs for many years, and it is recommended for patients with more aggressive disease [81]. There is also some evidence suggesting that chemotherapy may have more of a role for patients with pancreatic NETs (vs. non-pancreatic) [82]. The backbone of NET chemotherapy contains alkylating agents (streptozotocin (STZ), temozolomide) and fluoropyrimidines (5-fluorouracil (5-FU), capecitabine) [28,29,30,83,84,85]. Although single-agent schedules have been evaluated [86], combinations are preferred with response rates varying between 36–56% across the studies [28,29,30,83,84,85]. A summary of the main studies exploring the role of chemotherapy in patients with PanNETs is provided in Appendix A.

Unfortunately, most of the studies of chemotherapy in patients with PanNETs are retrospective series or small phase II trials. The only phase III study, published in 1992, explored the efficacy of STZ + 5-FU, STZ + doxorubicin, or chlorozotocin alone [30]. Higher response rates were observed with STZ-containing schedules (45% and 69% vs. 30%, respectively). Triple combinations were also explored, combining STZ and 5-FU with either doxorubicin or cisplatin [84,85], but are not recommended due to toxicity and insufficient PFS gain [85]. The combination of oxaliplatin with 5-FU/capecitabine in a few small phase II studies in combination with bevacizumab resulted in RRs between 30–41% [87,88,89]. 

Temozolomide has attracted attention lately due to its oral formulation. It has been combined with thalidomide [90], antiangiogenic drugs such as bevacizumab [91] or targeted therapies such as everolimus [92]. The most explored chemotherapy combination is temozolomide and capecitabine (TemCap) [31,32,33,34]. The synergy of both drugs may be due to capecitabine depleting the enzyme O6-methylguanine DNA methyltransferase (MGMT) [93,94,95]. The E2211 study was the first prospective randomised phase II trial in chemotherapy-naïve patients diagnosed with advanced PanNETs, who were randomised to TemCap vs. temozolomide alone [31]. The study showed an improvement in PFS in favour of TemCap (median PFS 22.7 vs. 14.4 months), regardless of the tumour grade (*p*-value 0.410); of note there was a higher prevalence of G1 patients in the TemCap arm. Toxicity is mainly in the form of myelosuppression, which raises the issue of the optimal length of therapy [96]. 

### 2.5. Evidence Supporting the Use of Immunotherapy

The concept of immunotherapy is not new to NENs, since alpha interferon (IFN) has been used in the field for many years [97]. Recently, the SWOG S0518 phase III trial in advanced midgut NETs, failed to show superiority of IFN alpha-2b in combination with octreotide LAR 20mg, compared to bevacizumab plus SSAs with similar, although long, median PFS (15.4 vs. 16.6 months) [98].

Unfortunately, immunotherapy with checkpoint inhibitors in patients with PanNETs has led to disappointing results (Appendix A). Pembrolizumab, an anti–programmed death-1 (PD-1) antibody, has been explored in well-differentiated NEN population first in a phase Ib study (KEYNOTE -028 [49]; 16 patients with PanNETs included) followed by a phase II clinical trial (KEYNOTE-158 [48]; number of patients with PanNETs not specified). Reported median PFS was short (4.5 months in PanNETs) [49] and observed responses were low (6% in PanNETs) [49]. The new anti-PD-1 antibody spartalizumab was tested in a phase II trial in a mixed population of patients with non-functioning NENs (well- and poorly-differentiated) and a variety of primaries, independent of programmed death-ligand 1 (PDL-1) status (33 were PanNETs) [50]. The responses were low, with a 3% RR in patients with PanNETs [50].

Althogh many options have been explored, current systemic therapeutics approved by EMA and FDA are limited to SSAs—Sandostatin LAR, Lanreotide Autogel; targeted therapy—sunitinib and everolimus; PRRT—177Lu-Dotatate; chemotherapy—temozolomide, capecitabine, streptozocine, dzcarbazine, 5-Fu and oxaliplatin; immunotherapy—interferon alpha 2b, Table 2.

## 3. Considerations for Treatment Selection

Figure 1 summarises the pooled median PFS (Figure 1A) and ORRs (Figure 1B) reported in the above-mentioned studies, by treatment group. 

Although patients with PanNETs were only included in one of the main randomized studies exploring the anti-proliferative effect of SSAs, it is clear that SSAs have a role for controlling tumour growth and managing symptoms in patients with functioning PanNETs; median PFS is high, despite a very low ORR. In contrast, the role of SSAs is probably limited for patients with non-functional PanNETs in the presence of more aggressive features, like Ki-67 > 10%, larger tumour burden, or the presence of cancer-related symptoms. 

Although many trials have explored multiple targeted therapies, only everolimus and sunitinib are currently approved and available for the management of patients with PanNETs. Their role seems to be independent of functional status, although there is a preference of everolimus for insulinomas due to its capacity to produce hyperglycaemia. Based on available data, targeted therapies are able to achieve prolonged disease control, despite low objective response rates. The only exception seems to be lenvatinib in the treatment of patients with advanced G1/G2 PanNETs and gastronintestinal NETs (achieved a high RR) [72], but its activity is still to be proven in the context of a phase III study. 

The role of PRRT in patients with PanNETs is currently open to discussion; no randomised phase III trial is available to support its use, which makes the interpretation of available phase II and retrospective study evidence challenging. When comparing the pooled PFS data (Figure 1A), it could be argued that PRRT provides the longest median PFS and should, therefore, be the first choice for patients with PanNETs. However, the data are to be interpreted with caution, since patient selection is likely to represent a very significant bias when comparing these studies. The fact that all patients treated with PRRT are required to be SSTR+ (likely to be a better prognosis population) must be taken into account and could explain why PRRT appears to have the longest median PFS [99,100,101].

The choice of chemotherapy in clinical practice in these patients seems to favour TemCap, a decision that was supported by its favourable toxicity profile and oral administration route. The RR is the highest of all therapy options (Figure 1B), despite this superiority not being translated into the longest PFS (likely to represent a selection bias with patients treated with TemCap having a higher Ki-67, tumour burden, and worse prognosis). Finally, further studies and possible combinations strategies are likely to be required prior to immunotherapy becoming part of the therapeutic arsenal for patients with PanNETs.

### 3.1. Building an Individualized Plan

Current ENETS guidelines recommend surgery (even in the presence of distant metastases) for patients with PanNETs if disease is fully resectable [15]. In patients with PanNETs, the predominant metastatic site is the liver, which accounts for 40–93% of all metastases [107]. There is evidence supporting the fact that resection of liver metastases improves survival [108,109] Positive results from the synchronous resection of primary and the liver metastases [110,111] were the basis of the latest ENETS recommendations [58]. Following curative surgery, there is no clear evidence supporting adjuvant strategies for PanNETs [112]. For patients with functioning tumours, supportive medications, such as diazoxide in insulinoma, PPI in gastrinoma and somatostatin analogues (octreotide LAR or lanreotide autogel), or IFN-alpha 2b are recommended, with options, such as debulking surgery or loco-regional or ablative therapies, used to treat refractory symptoms. Liver transplantation is an option for selected patients [15]. 

In the absence of surgical options, for patients with advanced disease diagnosed with non-functional PanNETs (G1 or G2 (Ki-67 < 10%)) who have low tumour burden and no symptoms and in the absence of documented disease progression, a watch-and-wait approach could be considered. This strategy has a more limited role in everyday practice for locally advanced or metastatic disease due to very well tolerated available therapeutic options and it should be considered in very selected patients. Alternatively, potential subsequent therapeutic strategies include SSAs, targeted therapies, chemotherapy, and PRRT. For patients with G2 non-functional PanNETs, high tumour burden, and/or documented progression or cancer-related symptoms a recommendation strategy could contain first line chemotherapy, followed by second-line targeted therapies and third line PRRT. Other guidelines, such as the North America Neuroendocrine Tumour Society (NANETS) updated in 2013 [113] and the National Comprehensive Cancer Network (NCCN) last updated in 2019, are more broad [114]; there is less focus on the tumour grade for treatment selection, but they all suggest watchful waiting as an option for patients with low-volume disease and the opportunity for initiation of targeted agents or even PRRT (NCCN) in the case of symptomatic or high volume disease. 

Treatment selection for advanced disease depends not only on tumour-specific characteristics, such as grade or Ki-67, but also on the aim of therapy and other factors (Figure 2), which will now be discussed. A multidisciplinary approach is crucial for the management of patient with rare diseases, like PanNETs, and referring patients to NEN-dedicated teams is recommended [115,116].(1).Pathology: tumours with low Ki-67 or low-grade are usually associated with higher expression of SSTR, a more indolent disease course and poorer responses to chemotherapy, in comparison with tumours with high Ki-67 or higher-grade [117]. Low Ki-67 seems to be an independent predictor of better response to PRRT [118], although the observed benefit was present in both G1 (HR 0.24 (95%-CI 0.13–0.44)) and G2 (HR 0.15 (95%-CI 0.07–0.34)) patients with midgut NETs [36]. Similarly, the benefit from lanreotide was comparable in both groups of patients [G1 (HR 0.43 (95%-CI 0.25–0.74)) vs. G2 (Ki-67 up to 10%) (HR 0.45 (95%-CI 0.22–0.91)]. Patients with PanNETs with a Ki-67 >10% were not included in the CLARINET study [19]; thus, evidence to support the use of SSAs in this patient group is scarce [119]. The phase III study exploring the role of sunitinib in the treatment of patients with PanNETs suggested an impact on PFS [120], more marked in the group of patients with Ki-67 < 5% (HR 0.38 (95%-CI 0.16–0.92)) [vs Ki-67 > 5% (HR 0.63 (95%-CI 0.24–1.71))] [34]; however, it has to be mentioned that the limited number of patients with Ki-67 data available in this study (36 patients out of the 86 patients in each arm), limited the power of this subgroup analysis. Everolimus showed similar PFS benefit in both groups: G1 (HR 0.41 (95% CI 0.31–0.53), G2 (HR 0.21 (95% CI 0.11–0.42)) [23].(2).Functional vs. Non-functional disease: functionality is associated with well differentiated tumours, lower grade, low Ki-67, and higher expression of SSTRs. For symptomatic control, SSAs or PRRT are considered. However, the PFS impact reported in the CLARINET study in patients with non-functioning PanNETs supports the use of lanreotide, regardless of functioning status [19,60]. No subgroup analysis by functional status was performed in the NETTER-1 study [36]. The non-functional population may benefit more (HR 0.26 (0.13–0.54)) from sunitinib therapy compared to the functional cohort (HR 0.75 (0.30–1.84)); acknowledging the limited number of patients (and therefore limited power) in this latest group (86 vs. 46 patients, respectively) [26]. Such information is not available for everolimus [23]. (3).Tumour load, liver involvement: large tumour burden as well as large liver tumour load and presence of extrahepatic metastases are negative prognostic features in patients with PanNETs [10]. However, benefits from SSAs and PRRT were seen in both the CLARINET and NETTER trials, with improvements in PFS, irrespective of tumour load or the presence of extrahepatic disease [19,36]. In addition, the benefit from sunitinib in the landmark trial was seen regardless of number of sites of disease (≤2 or ≥3) [121]. Despite this similar effect being reported in clinical trials, for patients with higher tumour load, the aim of therapy may be in favour of reducing tumour burden, for which strategies with higher objective response rates may be considered. These could be in the form of chemotherapy or PRRT, while SSAs and targeted therapies (reported to achieve lower response rate) may be reserved for patients with lower tumour burden. One of the exceptions may be lenvatinib, which leads to high response rates [72] and may change the management approach, if the results are confirmed in phase III studies. The type of distant metastases also represents the aggressiveness of the disease and it is also illustrated in a recently published analysis of SEER data from 2010 to 2014 [122]. Involvement of liver and bone is connected with worse prognosis followed by brain metastases (not very common in PanNETs). Scoring systems to assign groups with different survivals may also be useful in calculating the individual risk in different NENs [122]. Tumour growth rate (TGR) is another factor that is able to predict PFS and response to treatment [123].(4).Other individual factors may be considered when selecting therapy. A watch-and-wait approach could be a suitable strategy in selected patients with grade 1 PanNETs when comorbidities or other patient-related factors may play a significant role. In addition, selection of a specific targeted therapy (i.e., sunitinib vs. everolimus) may be supported, depending on specific comorbidities: favouring everolimus for patients with hypertension or in the scenario of an insulinoma and favouring sunitinib for patients with diabetes or past medical history of lung comorbidities. The presence of MGMT deficiency has been suggested as a predictive marker for temozolomide, even though its role as a predictive factor in patients with PanNETs is still to be clarified, and testing should not be employed in routine clinical practice [94,124]. Similarly, the predictive role of PD-L1 as a biomarker for immunotherapy treatment has been suggested in other disease groups [125,126,127]. However, despite 42% of patients with PanNETs being expected to express PD-L1 [128], the lack of responses in studies using immunotherapy [48,49,50] indicates that this biomarker may not be relevant in NENs. (5).Prior therapies administered may impact future therapy options; both sunitinib and everolimus showed efficacy in the setting of treatment-naïve and pre-treated (including SSAs or chemotherapy) patients [22,24,27,65]. In contrast, the CLARINET study only included treatment-naïve patients and therefore the use of SSAs for patients who had already progressed on previous treatments is unclear [19]. Unfortunately, use of PRRT frontline or as a subsequent line of therapy could not be definitely settle down based on the study by Brabander and colleagues, where no baseline characteristics for the studied PanNETs population was provided [80]. Chemotherapy has been used in first/subsequent lines of therapies in PanNET patients and benefit has been shown, regardless of prior treatment. This could be illustrated in the E2211 trial, which included patients with advanced PanNET who were randomized to receive Temozolomide or TemCap combination. The fact that almost half of patients in the E2211 trial received prior targeted therapies and that 53% of patients randomised to TemCap received concomitant SSAs supports the role of chemotherapy, regardless of the line of therapy [31].

### 3.2. Challenges in the Sequencing of Therapy

With the expansion of available therapeutic options, the concept of planning a “continuum of care” is becoming more relevant. This approach includes, “individualised planning, in which patients are given the opportunity to benefit from exposure to all active agents and modalities, while minimising unnecessary treatment and toxicity, with the ultimate goal of improving survival as well as quality of life” [129]. Therefore, one of the main challenges that clinicians currently face in treating patients with advanced PanNETs is the difficulty of selecting not only the first-line, but also subsequent lines of therapy. It is most likely that one specific sequencing strategy will not be applicable to all patients. While the guidelines provide a simplified framework, the reality is that individual characteristics (e.g., performance status, type of progression—oligo, systemic, comorbidities, disease features (grade, Ki-67, functional status), previous treatment regiments, and the response to them as well as side effects) at the time of each change in line of therapy are to be revisited, within a multidisciplinary setting, in order to provide patients with the best treatment sequence strategy. 

Neoadjvant approach in locally advanced or even metastatic PanNETs with surgical treatment in responders to first line systemic therapy is a potential strategy [130]. In the recent years, some studies report good results and R0 resections after neoadjuvant PPRT [131,132,133,134]; however, this should be validated in larger studies and patient selection has to be improved.

Although some ongoing clinical trials (i.e., SEQTOR [135] or COMPETE [136]) may answer specific questions regarding sequencing between two therapies (chemotherapy vs. targeted (SEQTOR); targeted vs. PRRT (COMPETE)), they are unlikely to answer all of the questions that will arise in clinical practice, in view of the high number of potential sequencing combinations if all therapy options are to be taken into account (Figure 3).

Following progression to SSA, whether to maintain SSA or not while administering other concomitant systemic therapies is debatable. In functioning disease, controlling the symptoms may be a reason to continue SSAs. However, the majority of PanNETs are non-functioning and the survival advantage in prolonging SSA application is not yet clear. In combination with PRRT, for example, providing more strong SSRT blockade may translate in longer PFS and OS [137]. Synergetic mechanism of action on PI3K/AKT/mTOR pathway [138] is one probable reason of combing SSAs and everolimus that could be seen in the result from RADIANT-1 confirmed in phase II EVERLAR study [22,139]. There were some promising results regarding RR especially in PanNETs from couple of trials exploring the combination of metronomic chemotherapy (cepecitabine or temozolomide) with Bevacizumab and SSAs [140,141]. In the study with Temozolomide induced depletion of MGMT by prolonged administration of the cytotoxic drug is discussed to be potential reason for the observed responses, but this needs to be further explored [140].

## 4. Ongoing Challenges

The field is currently facing multiple challenges. Firstly, the assessment of tumour response to therapy remains challenging in NETs. Significant improvement in response assessment in patients with PanNETs have been made, moving away from using clinical/biological responses to the use of validated WHO (World Health Organization) and RECIST (Response Evaluation Criteria in Solid Tumours) criteria [142]. In addition, more sensitive and specific nuclear imaging techniques visualising SSTR expression (111In- SPECT, 68-GaPET/CT) may be used for the diagnosis and assessing response to therapy beyond RECIST. Combining different imaging technologies and assessment of disease with cross-sectional and molecular imaging is increasingly relevant for patients with PanNETs, not only for treatment selection following a theranostic approach, but also for assessment of response to therapy [143,144]. The development of novel radiological assessment strategies may also be of help (i.e., alternative cut-off definitions for assessment of response [145] or assessment of Tumour Growth Rate (TGR) [123,146,147].

Secondly, the development of biomarkers remains an unmet need. The currently used Chromogranin-A and 5-hydroxyindoleacetic acid (5-HIAA) are known to have limitations with false positive and negative results. However, there is a lack of alternatives beyond the use of fasting gut hormones. Alternative possible biomarkers in the form of growth factors, such as VEGF and its receptors, placental growth factor (PlGF), Interleukin-8 (IL-8), PD-1/PD-L1 expression, T-cell immunoglobulin and mucin domain 3 (TIM3), X-linked transcriptional regulator (ATRX), death domain-associated protein 6 (DAXX) genes, circulating tumour cells, and circulating tumour DNA, have been studied without consensus on their utility [148,149,150,151,152]. The use of biomarkers can be employed beyond the selection of patients for specific therapies; understanding the biology of PanNETs, especially the mechanisms of acquired resistance (for example, to targeted therapy, although this applies equally to other treatments) is needed. The impact of previous treatment on subsequent treatment efficacy is also an ongoing research question. 

Thirdly, in view of increasing treatment options, cost-effectiveness is an important aspect to consider. Some initial attempts comparing everolimus, ^177^Lu PRRT and sunitinib in advanced, unresectable, or metastatic progressive NETs have been reported [153]. However, information coming from such studies has to be interpreted carefully and its implementation into clinical practice may be challenging. 

## 5. Future Perspectives 

The future of systemic treatment for patients with advanced PanNETs may rely on therapy combinations. While monotherapy strategies have already been shown to be effective, there is limited information regarding potential combinations with the aim of pursuing synergistic effects. Thus, the ongoing trend of combining therapeutic approaches in clinical trials with different mechanisms of action to gain more advantages in effect and in overcoming resistance pathways may be a future therapeutic option for patients with PanNETs (Figure 4). 

It is also worth highlighting the significant change in the design of ongoing clinical trials in the field (Figure 5). While a few years ago (Figure 5A, most combination strategies were exploring the role of targeted agents combined with other molecules, the current scenario seems very different, with most combination studies exploring the addition of chemotherapy (mainly based on alkylating agents) to other therapeutics strategies (Figure 5B).

Expanded research is needed in order to evaluate the role of SSA beyond its role as a first-line option for PanNETs. The current evidence relies on the CLARINET study, which was not powered for subgroup analysis; with no level-1 evidence to support the use of octreotide as an anti-proliferative in patients with PanNETs. The role of SSAs as maintenance therapy (REMINET study; NCT02288377) and as a drug to combine with other therapeutic options (e.g., targeted therapies (everolimus/metformin (MetNET1, NCT02294006); everolimus +/− bevacizumab, NCT01229943) or chemotherapy (capecitabine/bevacizumab (XELBEVOCT, NCT01203306)) and the role of high dose SSAs (CLARINET-Forte, NCT02651987) are still to be determined. 

Additional data supporting the role of PRRT in patients with PanNETs would strengthen the evidence base, given the limitations of the Brabander study [80]. The power of the study cannot be assessed, as the study was not randomised-controlled (which makes it difficult to clarify if the prolonged PFS is due to PRRT or due to patient selection) and did not provide a sample size calculation [80]. The use of radioisotope treatment in the neoadjuvant [131,132,133,134] or adjuvant setting [41,154], alone or in combinations with other therapeutics [130], is being investigated. However, future prospective randomised trials exploring PRRT usage need to provide clarity regarding patient selection, including predictive biomarkers of treatment response (beyond SSTR expression). Novel radioisotopes are being investigated as well as novel targets, with some promising results reported in studies while using SSTR antagonists (which block more SSTR2 receptors [136,155]). Building a more personalised approach by improving the dosimetry techniques for pre-therapeutic measurement of organ and tumour exposure may provide optimal radionuclide therapy for each patient, accounting for individual characteristics (disease, age and comorbidities: “personalised dosimetry” [156]).

Diverse objective responses achieved with targeted therapies are poorly understood. For many years targeted therapies achieved low RRs. However, lenvatinib has achieved a higher RR in patients with PanNETs, but the mechanism is still not clear [72]. Similarly, phase IV data from the use of sunitinib in PanNETs have also reported higher RRs than previous studies (9.3% in phase III study vs. 24.5% in phase IV study) [27,67]. Whether this is reflection of better patient selection, more adequate management of toxicities, with increased exposure to active drugs, or evolution of imaging techniques can only be speculated. A better understanding of these factors is required when considering targeted therapies in the neoadjuvant setting, especially in cases where good responses are observed. 

Finally, the role of immunotherapy in patients with PanNETs will require further research and “outside-the-box” thinking following initial disappointing results. Strategies to convert immune “cold” tumours into “hot” are probably required in order to move this research area forward [157,158].

## 6. Conclusions 

The management of patients with advanced PanNETs is becoming increasingly complex due to expanding treatment options. The main goal is the construction of an individualised treatment strategy that is based on multidisciplinary discussion, not only at first presentation but also when changing lines of therapy. Treatment decisions should be based on patient-related and tumour-related factors and the need for cytoreductive or stabilisation effect. Research regarding choices of different therapeutic strategies, including SSAs, targeted therapy, PRRT, chemotherapy, and/or immunotherapy in combination or sequentially in the treatment of patients with advanced PanNET is needed, as is the development of novel prognostic and predictive biomarkers. 

## Figures and Tables

**Figure 1 cancers-12-01988-f001:**
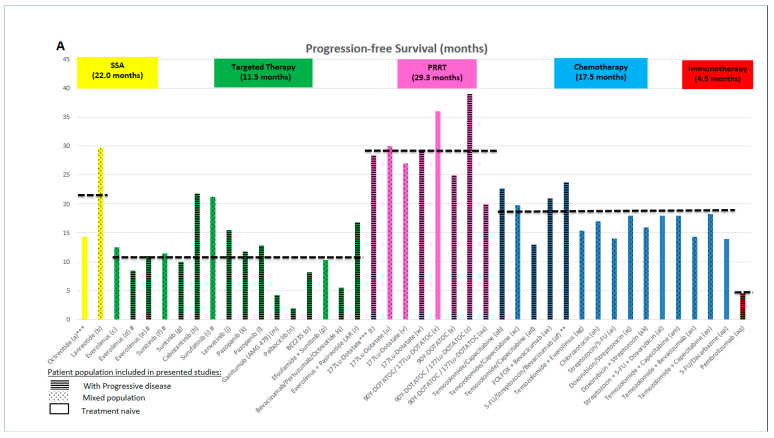
(**A**) Graphical representation and pooled progression-free survival (PFS) and (**B**) objective response rate (ORR) by treatment group. Data for the studies summarised in Appendix A are presented. Mean PFS and ORR are calculated from studies with data available. PRRT: Peptide receptor radionuclide therapy; SSA: somatostatin analogue. a—PROMID trial—Octreotide arm [20,21]; b—CLARINET trial—Lanreotide arm [19,60]; c—Phase II study, Yao et al., 2008—Everolimus arm [102]; d—RADIANT-1—Everolimus arm [22]; e—RADIANT-3—Everolimus arm [24,103]; f—SUN111—Sunitinib arm [26,27]; g—Retrospective, Rinzivillo et al., 2018—Sunitinib arm [65]; h—Phase II study NCT01466036—Cabozantinib arm [71]; i—Phase I/II study NCT02267967—Surufatinib arm [73]; j—TALENT trial—Lenvatinib arm [72]; k—Phase II study, Phan et al., 2010—Pazopanib arm [18]; l—PAZONET—Pazopanib arm [70]; m—Phase II study NCT01024387—Ganitumab (AMG 479) arm [77]; n—PALBONET—Palbociclib arm [74]; o—Phase II study, Salazar et al., 2018—BEZ235 arm [63]; p—SUNEVO (GETNE 1408)—Efosfamide/Sunitinib arm [104]; q—Phase II study, Bendell et al., 2016—Bevacizumab/Pertuzumab/Octreotide arm [105]; r—COOPERATE-2 trial—Everolimus/Pasireotide LAR arm [62]; t—NETTER-1 trial—177Lu-Dotatate arm [36,106]; u—Retrospective, T.Brabader et al., 2017—177Lu-Dotatate arm [37]; v—Prospective observational study, Garske-Román, U et al., 2018—177Lu-Dotatate arm [38,43]; w—Phase II study, Sansovini et al., 2017—177Lu-Dotatate arm [39,43]; x—Prospective trial, Bertani et al., 2016—90Y-DOTATOC/177Lu-DOTATOC arm [41,43]; y—Phase II study, Rogowski et al., 2016—90Y-DOTATOC arm [43,44]; z—Retrospective, Horsch et al., 2016—90Y-DOTATOC/177Lu-DOTATOC arm [43,47]; aa—Retrospective, Baum et al., 2018—90Y-DOTATOC/177Lu-DOTATOC arm [43,46]; ab—E2211 trial—Temozolomide/Capecitabine arm [31]; ac—Retrospective study, De Mestier et al., 2019—Temozolomide/Capecitabine arm [32]; ad—Retrospective study, Campana et al., 2018—Temozolomide/Capecitabine arm [33]; ae—Phase II study, Kunz et al., 2016—FOLFOX + Bevacizumab arm [89]; af—BETTER trial—5-FU/Streptozocin/Bevacizumab arm [28]; ag—Phase I/II, Chan et al., 2013—Temozolomide + Everolimus arm [92]; ah—Phase III study, Moertel et al., 1992—Chlorzotocin-arm [30]; ai—Phase III study, Moertel et al., 1992—Streptozotocin/5-FU arm [30]; aj—Phase III study-Moertel et al., 1992—Doxorubicin/Streptozotocin arm [30]; ak—Retrospective study, Delaunoit et al., 2004—Doxorubicin + Streptozotocin [83]; al—Retrospective, Kouvaraki et al., 2004—Streptozotocin/5-FU/Doxorubicin arm [84]; am—Retrospective, Strosberg et al., 2011—TEM/CAP arm [34]; an—Phase II study, Chan et al., 2012—Temozolomide/Bevacizumab arm [91]; ao—Retrospective, De Mestier et al., 2019—EM/CAP arm [35]; ap—Retrospective, De Mestier et al., 2019—5-FU/Dacarbazine arm [35]; aq—KEYNOTE -028—Pembolizumab arm [49]; ar—Single-arm open-label study, Halperin et al., 2019—Ziv-Aflibercept arm [75]; as—Phase II study, Jin et al., 2016—Panobinostat arm [76]; at—Phase II, Imhof et al., 2011—90Y-DOTATOC arm [42]; au—Expanded access trial, Hamiditaba et al.r, 2017—177Lu-DOTATOC arm [45]; av—Dumont et al., 2015—90Y-DOTATOC/177Lu-DOTATOC arm [40]; aw—Retrospective study, Turner, 2010—Streptozotocin/5-FU/Cisplatin arm [85]; ax—Phase II study, Ramanathan, 2001—Dacarbazine arm [86]; ay—Phase II study, Kulke, 2006—Temozolomide/Thalidomide arm [90]; az—Phase II study, Venook, 2008—5-Fu/Oxaliplatin/Bevacizumab arm [87]; ba—Phase II study, Kunz, 2010—Capecitabine/Oxaliplatin/Bevacizumab arm [88]; bb—Phase II multicentre study, NCT02955069—Spartalizumab arm [50]; * No patients with PanNETs were included in the trial; ** at 24th month; *** NO SEPARATE INFORMATION ABOUT patients with PanNETS; # Central review.

**Figure 2 cancers-12-01988-f002:**
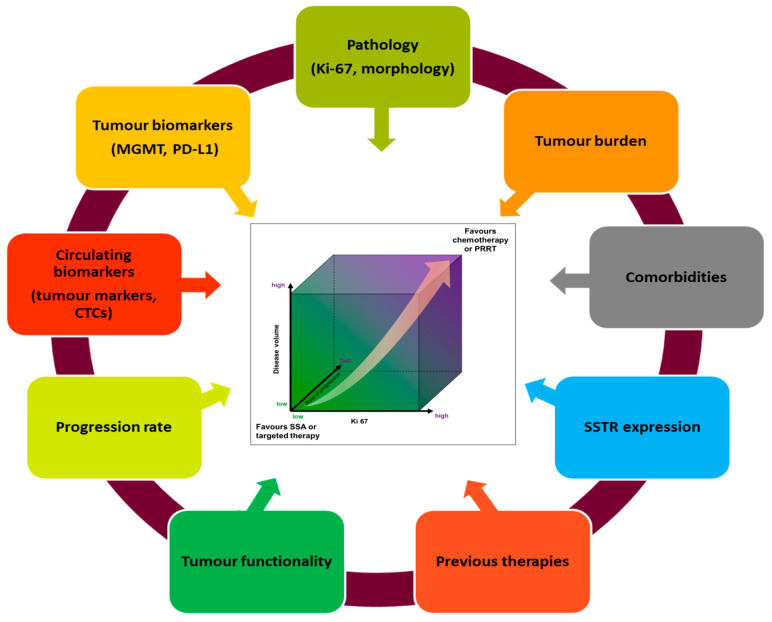
Summary of factors to consider at the time of treatment decision for patients with advanced PanNETs suitable for systemic therapies. Figure adapted from [81]. MGMT—O6-methylguanine DNA methyltransferase; PD-L1—programmed death-ligand 1; SSTR—somatostatin receptor; and, CTCs—circulating tumour cells.

**Figure 3 cancers-12-01988-f003:**
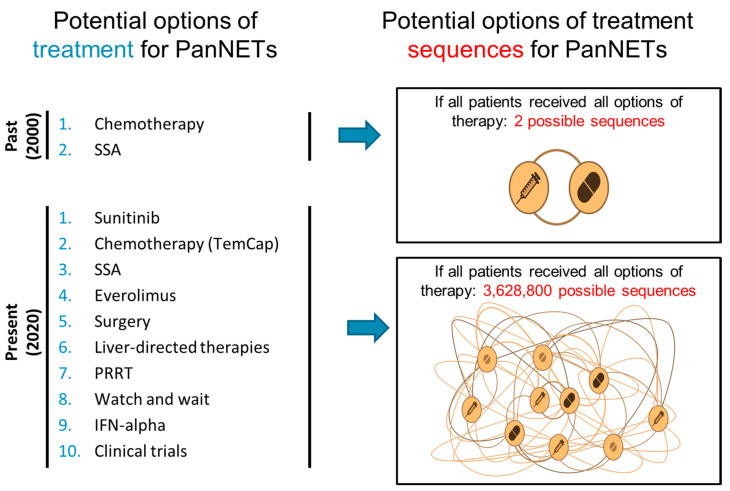
As options of treatment increase, potential sequencing alternatives grow exponentially making it difficult for clinical trials to explore each potential sequencing option. PanNETs—pancreatic neuroendocrine tumours; SSA—somatostatin analogue; TemCap—temozolomide and capecitabine; and, PRRT—Peptide receptor radionuclide therapy IFN- interferon.

**Figure 4 cancers-12-01988-f004:**
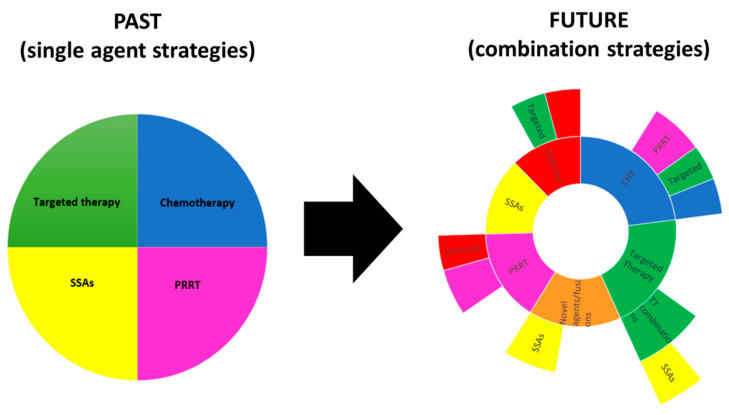
Comparison between previously used and currently explored treatment options. On the left you can see used before main therapeutic options focused on single treatment modality and on the right there is a graphic visualisation of currently explored treatment options expanding beyond monotherapies and exploring advantages of combining therapeutics with different mechanism of action. CHT—Chemotherapy; PRRT—Peptide receptor radionuclide therapy; SSAs—somatostatin analogues; IMT—immunotherapy; and, TT—targeted therapy.

**Figure 5 cancers-12-01988-f005:**
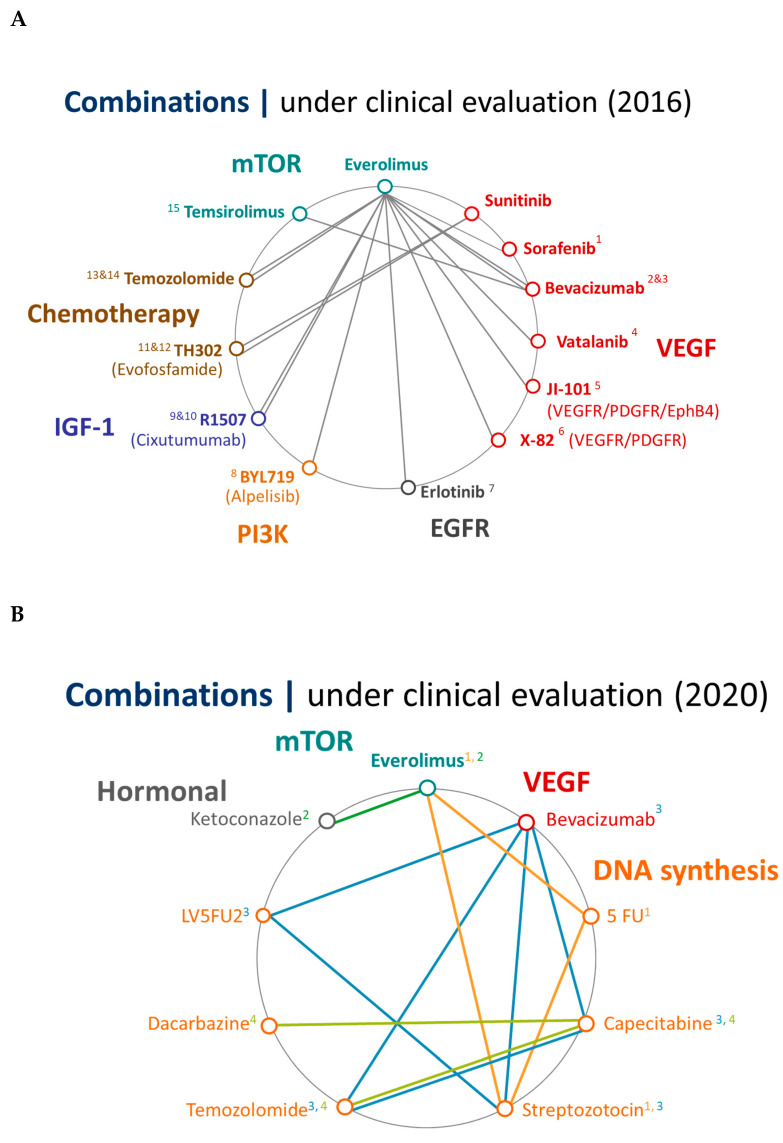
Ongoing combination trials in patients with PanNETs. Figure 5B: Combinations | under clinical evaluation (2020). Figure 5A: Combinations | under clinical evaluation (2016). (**A**) ^1^NTC00942682 (Phase I), ^2^NCT00607113 (Phase II), ^3^NCT01229943 (randomised Phase II), ^4^NCT00655655 (Phase I), ^5^NCT01149434 (Phase I/II), ^6^NCT01784861 (Phase I/II), ^7^NCT00843531 (Phase II), ^8^NCT02077933 (Phase I), ^9^NCT00985374 (Phase I/II), ^10^NCT01204476 (Phase I), ^11^NCT01381822 (Phase I), ^12^NCT02402062 (Phase II), ^13^NCT00576680 (Phase I/II) Grade1&2, ^14^NCT02248012 (Phase II) Grade3, ^15^NCT010126 (Phase II). (**B**) 5FU: 5-fluoro uracil, LV5FU2: folinic acid. 5FU (V bolus & 48 h continuous infusion). ^1^NCT02246127 (SEQTOR, Phase III), ^2^NCT01263353 (COOPERATE-1, Phase I), ^3^NTC03351296 (BETTER 2, Phase II), ^4^NTC03279601 (Phase II).

**Table 1 cancers-12-01988-t001:** WHO Classification Pancreatic neuroendocrine tumours (PanNET), 2017.

Tumour Features	G1 NET	G2 NET	G3 NET	NEC *
Ki-67	<3	3–20	>20	>20
Mitosis	<2	2–20	>20	>20
Differentiation	Well differentiated	Well differentiated	Well differentiated	Poorly differentiated

* Large cell and small cell neuroendocrine carcinoma should be considered as separate subtypes. Legend: PanNETs—pancreatic neuroendocrine tumours; G—grade; NET—neuroendocrine tumour; NEC—neuroendocrine carcinoma.

**Table 2 cancers-12-01988-t002:** Approved systemic treatment options for PanNET by FDA and EMA.

Approved Treatments	EMA	FDA
Sandostatin LAR	V	V
Lanreotide Autogel	V	V
Pasireotide LAR	-	-
Sunitinib	V	V
Everolimus	V	V
Pazopanib	-	-
Surufatinib	-	-
Cabozantinib	-	-
Lenvatinib	-	-
Efosfamide +/− Sunitinib	-	-
Palbociclib	-	-
Ziv-Aflibercept	-	-
Bevacizumab +/− Pertuzumab	-	-
BEZ235	-	-
Panobinostat	-	-
Ganitumab	-	-
177Lu-Dotatate	V	V
90Y-DOTATOC	-	-
90Y-DOTATOC plus 177Lu-DOTATOC	-	-
177Lu-Oxodotreotide	-	-
Temozolomide	V	V
Capecitabine	V	V
Streptozocine	V	V
Dacarbazine	V	V
5-Fu	V	V
Oxaliplatin	V	V
Pembrolizumab	-	-
Spartalizumab	-	-
Interferon alpha 2b	V	V

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
