# Peer review of "Systemic Treatment Selection for Patients with Advanced Pancreatic Neuroendocrine Tumours (PanNETs)"

_cancers, 2020, doi:10.3390/cancers12071988_

Round 1

Reviewer 1 Report

This is a comprehensive, well-written review covering therapeutic approaches in patients with panNETs. The manuscript is clear, and is of potential interest for readers facing with these rare and heterogeneous disease

I only have some minor criticisms, which I would like to be addressed before considering this work suitable for publication.  

  • Targeted therapies chapter: I would add in the literature analysis also some available studies performed in the real-world setting, beyond phase-3 RCTs
  • I would appreciate a table summarizing which treatments are approved by regulatory agencies (FDA/EMA). This would give an additional information in the daily clinical practice for physicians dealing with panNETs
  • Figure 1A. Specify whether patients were progressive or not in each reported trial. Suggest to use a different graphical form for trial including progressive patients (i.e. red border of the column, or something similar). This could help the reader to easily understand differences among PFS
  • Give a brief comment on the role of combined therapies based on SSA (SSA + target, SAA+PRRT, SSA+chemotherapy), and the potential use (if any...) to maintain the analog when moving to second-line therapy.
  • Discussion to figure 2. When approaching the "tumor load" issue, a comment on the role of specific distant metastatic sites should be added (i.e. extra-hepatic lesions, bone mets...)
  • Same discussion, point 4. I agree with authors that "watch and wait" policy could be considered in case of comorbidities. I would highlight that, to date, this is probably the only reason not to treat advanced panNETs. In addition, it should be addressed that this is a very unlikely scenario, given the extremely good safety profile of SSAs. So, why not to treat (see the results from the Clarinet study regarding "stable" disease)?
  • Same discussion. Consider as additional factor when approaching therapeutic sequence slope of tumor growth before commencing treatment (if known) and FDG-PET finding
  • Figure 3. Watch and wait approach, as well as IFN should be reported as separate options, given the extremely low evidence supporting these strategies
  • In the conclusions, I would highlight the crucial role of multidisciplinary team to plan optimal therapetucih strategies for panNET patients

Author Response

Dear reviewer,

Thank you very much for your comments and please see the attachment with our responses to them.

Kindest regards

Vera Megdanova (on behalf of all authors)

Reviewer 2 Report

The authors performed a broad literature review to summarize treatments for advanced PanNETS. However, this report reflects several limitations:

  • the study does not represent a systematic review of the literature
  • a clear definition of "advanced" PanNETs was not stated
  • as expected from the title, the paper should be focused on advanced PanNETs, but this was not clarified in the aims, nor it is clear from the text, that represents a summary of all available treatments without a distinction. An algorithm to help clinicians was not considered. 
  • Surgery, which represents a very important point in the treatment selection strategy also of advanced PanNETs, is just mentioned
  • the cited studies are reported in the text up and forth in a confusing way
  • Supplementary tables are difficult to read

On the whole, this study is an extensive summary of a very complex matter, however, it does not clarify or help clinicians in the treatment decision-making better than the available guidelines, neither bring additional contributions into the field.

Author Response

(The authors gave the same response as above.)

Reviewer 3 Report

This review paper summarises the current state of play regarding therapeutic strategies for patients with PanNETs. SSA's, targeted therapies, PRRT, chemotherapy and immunotherapy are all covered. I congratulate the authors on a well structured and written review that successfully encompasses the existing body of literature. This is no easy task given the heterogeneity of the disease of interest and the multitude of existing and proposed therapies that are at hand. In this regard, the authors have done an admirable job of crystallising where we are in 2020 and beyond this, have attempted to aid the interested reader by discussing patient/treatment selection paradigms. Whilst I commend the authors on this work I do have some additional areas of consideration.

  1. neoadjuvant therapy: in an age of proven benefit of neoadjuvant therapy for all types of malignant disease, I would like to see the authors discuss this in more depth.
  2. molecular insights: our understanding of the genetic/molecular basis of disease has evolved exponentially over the last decade or so and this is beginning to drive personalised therapy. Can patient selection and specific therapeutic strategy be guided by this in PanNET? If the evidence for current treatments is lacking, are there promising advances that the reader should be aware of?
  3. I was pleased to see some attempt to reflect on the exhaustive review of literature by developing discussion on treatment selection. I would however like to see this developed further with proposed selection/management algorithms; this would set the scene for future prospective validation studies to follow. 

Author Response

(The authors gave the same response as above.)

Round 2

Reviewer 1 Report

Excellent work, I have no further comment to make.

Reviewer 2 Report

This is an extensive review of the literature about this topic. However, the text is difficult to follow in several Points and the paper do not a contribution additional to the existing Guidelines.

Here my comments:

  • Regarding the different treatments categories, the authors are going back and forward throughout the text adding anytime new information about the different Trials. This is very confusing. The authors should provide a conclusive paragraph for each treatment category. Also a table to summary advantages and disadvantages for each category could help 
  • Line 261: These sentences should represent a separate Paragraph
  • Line 264: Table 3 here described should be table 2
  • Neoadjuvant treatments were only breafly described
  • The sentence at line 357-358 should be placed elsewhere or deleted.